# A Decoupled Calibration Method Based on the Multi-Output Support Vector Regression Algorithm for Three-Dimensional Electric-Field Sensors

**DOI:** 10.3390/s21248196

**Published:** 2021-12-08

**Authors:** Wei Zhao, Zhizhong Li, Haitao Zhang, Yuan Yuan, Ziwei Zhao

**Affiliations:** National Defense Engineering College, Army Engineering University of PLA, Nanjing 210007, China; dnallzw@163.com (W.Z.); lizz0607@163.com (Z.L.); Y849251925@163.com (Y.Y.); ziiwei@163.com (Z.Z.)

**Keywords:** three-dimensional electric-field sensor, decoupled calibration, inverse matrix, multi-output support vector regression, ν-SVR

## Abstract

Aiming at the problem that the measured accuracy of the electric field intensity which is affected by the coupling interference by sensor output signal from the component of a three dimensional electric field, the causes of the coupling error was analyzed, and a decoupled calibration method based on support vector regression algorithm for three-dimensional electric field sensor is proposed. The solution of the decoupled calibration matrix was regarded as a multi-objective optimization process, and the optimal decoupling calibration matrix was obtained by the ν-SVR algorithm. The complex inverse calculation of the matrix was avoided, and the calculation error was reduced. A rotary calibration device was designed to accurately measure the angle between the induction electrode of the sensor and the electric-field vector, and an accurate calculation model of the theoretical electric field was established. The experimental results showed that the error between the calculated and theoretical values of the electric-field components obtained by the proposed method were smaller than those obtained by the traditional inverse matrix calibration method, the accuracy of the calibration was improved, the rationality of the calibration method was proven, and the accuracy of the three-dimensional electric-field intensity measurements was further improved.

## 1. Introduction

With the development of aerospace industry, lightning protection of aircraft has also become a research hot spot. In order to protect the aircraft from the interference of strong atmospheric electric field and damage of lightning during tropospheric flight, Atmospheric electric field intensity has become one of the important parameters of aircraft flight [1]. Therefor three-dimensional electric-field detection technology has been widely used in aviation, aerospace, and power system technology [2,3]. A three dimensional (3D) electric field system for atmospheric electric field measurement was proposed by Xing, which is capable of three orthogonal directions in *X*, *Y*, *Z*, measuring [4]. At present, three-dimensional electric-field sensors are mostly distributed [5], airborne, and field-mill types [6]. Distributed airborne electric-field sensors have disadvantages such as large sizes and complex structures, and significant interference to the measured electric field can easily occur, affecting the accuracy of the measurement [7]. Three groups of sensitive electrodes arranged on three orthogonal sensitive surfaces have been adopted on a field-mill type three-dimensional electric-field sensor. This arrangement is smaller than the distributed airborne field model, and it has been widely used in the field of three-dimensional electric-field detection [8]. Model, design and testing of field-mill electric field sensors to measure the electric field was reported [9], a reduced scale calibration device was designed, and a finite element analysis method was proposed to establish the three dimensional model of the electric field sensor calibration device by Cui [10]. However, in practice, due to the complex structure of the field-mill-type three-dimensional electric-field sensor, it is impossible to determine the quantitative information affecting the measurement accuracy by means of simulation technology. Therefore, calibration is still an effective technical means to solve or eliminate such influence.

The commonly used field-mill-type three-dimensional electric-field sensor contains three mutually orthogonal (*X*, *Y*, and *Z*) induction electrodes, as shown in Figure 1. In the actual measurement process, the voltage signal generated by the induction electrode is not only affected by the component of the electric field perpendicular to the electrode, but it is also affected by two other orthogonal components of the electric field. This phenomenon is called interaxial coupling. Essentially, due to the influence of the sensor structure, the accuracy of three-dimensional electric-field measurements is affected by the distortion of the electric field near the sensor directly. The power line distribution diagram and potential cloud diagram of an electric-field sensor placed in a uniform-intensity electric field are shown in Figure 2.

To reduce the coupling effect on the measurement accuracy, Mohammed presented a dual-axis accelerometer with a zero cross-axis sensitivity. The inter-axial coupling sensitivity coefficient represents the ratio between the sensitivity perpendicular to the measured axial direction and the axial direction, and the larger the ratio is, the greater the inter-axial coupling effect will be [11]. A highly sensitive resonant electric-field micro-sensor based on SOI technology was proposed by Yang [12]. To improve the electric-field coupling effect, a coplanar shutter electrode was adopted on the micro-sensor.

Calibration methods based on inverse matrix operations are widely used for the calibration of electric-field sensors. The matrix relating the output signal of the sensor and the actual electric-field component was established [7], and the decoupling calibration of the three-dimensional electric-field sensor was realized by solving the pseudo-inverse matrix. However, this calibration process was complicated and costly. A single-chip micro-electromechanical system (MEMS) three-dimensional electric-field sensor with axisymmetric induction electrodes was studied [13], and by using a differential circuit, the influence of coupling interference between the axes was reduced. However, because the induction electrode was integrated into a single chip, the structure of the chip was complex, there was an axial direction that could not be designed as a symmetric structure, and the coupling interference could not be effectively eliminated. A novel three-dimensional electric-field sensor based on a flexible substrate and a three-dimensional assembly method was proposed [14]. The coupling of the measured electric field in space was studied, a decoupling method for solving the generalized inverse matrix was adopted, and accurate measurement of the three spatial components of the electric field was realized. An inverse matrix operation method was proposed [15], the coupling sensitivity matrix of the sensor was obtained, and the calibration of the sensor was completed. However, in this calibration process, a specific sensor calibration position was necessary, and less test data was collected. Thus, the optimal sensitivity matrix could not be obtained, which resulted in large calibration errors. Aiming at the principle of three-dimensional sensor interdimensional coupling, a generalized inverse decoupling algorithm for solving the matrix based on a Newtonian cubic difference was proposed by Zhou [16], and the decoupling accuracy was further improved. This method is based on inverse matrix operations. When the matrix is singular or nearly singular in the operation process, the calculation is complicated and time-consuming, and there are calculation errors. In addition, due to the limitation of the amount of test data and errors of the calculation, this method cannot obtain the optimal decoupling calibration matrix.

The decoupling calibration matrix was considered to be a multi-objective optimization problem [17]. Each element of the matrix was set as an optimization target, and each element of the inverse matrix was obtained. On this basis, a decoupling calibration method for three-dimensional electric-field sensors based on the genetic algorithm was proposed. The optimal decoupling calibration matrix was obtained by setting the fitness function and genetic operator. The complex matrix inverse calculation was avoided, and the calculation error was reduced. However, the algorithm itself could easily fall into local optimal solutions. In the research of multi-objective optimization problems, the genetic algorithm was used to optimize the parameters of the weighted matrix and improve the design efficiency and control performance of a linear–quadratic regulator (LQR) controller [18]. In the coding process of the genetic algorithm, binary encoding was adopted, which was a decoding and slow optimization process. A calibration model with inter-dimensional coupling and angular deviation is established under a three-dimensional adjustable external electric field by Wu [19], and the method of solving the sensitivity coefficient matrix based on the differential evolution algorithm under this model is proposed. Accurate and stable are improved by using the algorithm, but this method is easy to fall into the local optimal solution too.

The input of the three-dimensional electric-field sensor is the signal of the electric-field intensity, and the output is the voltage value obtained by the three induction electrodes. Decoupling is a process of determining the conversion matrix between the input electric-field intensity and the three output voltage signals of the induction electrode. It is easy to form an ill-conditioned matrix when solving generalized inverse matrix problems, which leads to lower decoupling accuracy and even incorrect decoupling results. When the genetic algorithm is used for decoupling, it easily falls into local optimal solutions because of the great uncertainty in the selection of the algorithm parameters. Support vector regression (SVR) is a machine learning method developed on the basis of statistical theory and optimization theory [20], SVR is mainly used in pattern recognition, regression estimation, probability density function estimation and so on [21], which can minimize the structural risk and obtain the global optimal solution [22,23]; the problem of overfitting was avoided. On this basis, a three-dimensional electric-field-sensor decoupling method based on the multi-output support vector regression algorithm is proposed. A rotating calibration device was designed to accurately measure the intersection angle between the sensor electrode and the spatial electric-field vector. An accurate model for the theoretical value of the electric field was established. The experimental results showed that the algorithm was stable and reliable, and the measurement accuracy of the sensor was improved significantly.

## 2. Coupling Calibration Principle of Three Dimensional Electric Field Sensor

Spatial electric fields usually exist as three-dimensional vectors. Three-dimensional electric fields in any direction of space can be decomposed into three mutually perpendicular electric-field components, which are expressed in a Cartesian coordinate system as follows:(1)$$E_{0}=E_{x}+E_{y}+E_{z}$$
where ${\vec{E}}_{0}$ is the electric-field intensity, and ${\vec{E}}_{X}$, ${\vec{E}}_{Y}$, and ${\vec{E}}_{Z}$ are the three orthogonal components of three-dimensional electric field in the local coordinate system (LCS). The intensity varies with the position of the sensor’s sensing electrode.

In the process of obtaining three-dimensional electric-field measurements, the electric-field intensity is reflected by the voltage output from the induction electrode of the electric-field sensor, and the relative position relationship between the three-dimensional electric-field vector and the sensor’s induction electrode is reflected by the voltage. The alternating current (AC) voltage signal output by any induction electrode of the three-dimensional electric-field sensor is affected by the three components of the three-dimensional electric field in the LCS coordinates. The relationships are expressed as follows:(2)$$\left\{\begin{array}{c} u_{XX}=k_{XX}\cdot E_{X}\\ u_{XY}=k_{XY}\cdot E_{Y}\\ u_{XZ}=k_{XZ}\cdot E_{X} \end{array}\right.$$
(3)$$\left\{\begin{array}{c} u_{YX}=k_{YX}\cdot E_{Y}\\ u_{YY}=k_{YY}\cdot E_{Y}\\ u_{YZ}=k_{YZ}\cdot E_{Y} \end{array}\right.$$
(4)$$\left\{\begin{array}{c} u_{ZX}=k_{ZX}\cdot E_{Z}\\ u_{ZY}=k_{ZY}\cdot E_{Z}\\ u_{ZZ}=k_{ZZ}\cdot E_{Z} \end{array}\right.$$
where $u_{XY}$ is the voltage and amplitude of the AC electrical signal generated by the *X*-direction induction electrode of the three-dimensional electric-field sensor under the influence of $E_{X}$, and $k_{XY}$ is the sensitivity coefficient of the *X*-direction induction electrode to the *Y*-direction electric-field component. The input and output model of three-dimensional electric field sensor is shown in Figure 3.

Linear Equations (2)–(4) can be written in a matrix form, as follows:(5)$$\left[\begin{array}{ccc} u_{XX} & u_{XY} & u_{XZ}\\ u_{YX} & u_{YY} & u_{YZ}\\ u_{ZX} & u_{ZY} & u_{ZZ} \end{array}\right]=\left[\begin{array}{ccc} k_{XX} & k_{XY} & k_{XZ}\\ k_{YX} & k_{YY} & k_{YZ}\\ k_{ZX} & k_{ZY} & k_{ZZ} \end{array}\right]\cdot \left[\begin{array}{ccc} E_{X} & & \\ & E_{Y} & \\ & & E_{Z} \end{array}\right]$$

The components of the three-dimensional (3D) electric-field sensor are coupled in reality. The electric-field component acting in a direction of an induction electrode that is perpendicular to the electric-field sensor has an impact on the output voltages of the induction electrodes in the other two directions. The measured voltage data of the induction electrodes are represented as follows:(6)$$\left\{\begin{array}{c} u_{X}=u_{XX}+u_{YX}+u_{ZX}\\ u_{Y}=u_{XY}+u_{YY}+u_{ZY}\\ u_{Z}=u_{XZ}+u_{YZ}+u_{ZZ} \end{array}\right.$$
where $u_{ii}$ is the electric-field intensity component in the direction that is perpendicular to the induction electrode, $u_{ii}$ affects the voltage output of the induction electrode, and the parameter $u_{i}{}_{j}$ is the electric-field component in another direction, which also affects the output voltage of the induction electrode. Thus, if a one-dimensional (1D) electric field is applied parallel to the *X*-direction, it will affect the voltage values of the induction electrodes in the *Y*- and *Z*-directions. Therefore, the voltage signals generated by the three induction electrodes can be expressed as follows:(7)$$\left(\begin{array}{c} u_{X}=k_{XX}\cdot E_{X}+k_{YX}\cdot E_{Y}+k_{ZX}\cdot E_{Z}\\ u_{Y}=k_{XY}\cdot E_{X}+k_{YY}\cdot E_{Y}+k_{ZY}\cdot E_{Z}\\ u_{Z}=k_{XZ}\cdot E_{X}+k_{YZ}\cdot E_{Y}+k_{ZZ}\cdot E_{Z} \end{array}\right.$$
voltage is a scalar parameter. The following relationship is obtained by analyzing Equation (7):(8)$$u_{i}=k_{Xi}\cdot E_{X}+k_{Yi}\cdot E_{Y}+k_{Zi}\cdot E_{Z}$$
where $u_{i}$ is the output voltage of the induction electrode *i* in the electric field, and $E_{X}$, $E_{Y}$, and $E_{Z}$ are the decomposed vectors of the electric field in the orthogonal *X*-, *Y*-, and *Z*-directions, respectively, as the sensor changed its position. The output voltage signals and the electric-field components of the 3D electric-field sensor with the three induction electrodes that were perpendicular to each other can be expressed as follows:(9)$$\left[\begin{array}{c} u_{X}\\ u_{Y}\\ u_{Z} \end{array}\right]=\left[\begin{array}{ccc} k_{XX} & k_{YX} & k_{ZX}\\ k_{XY} & k_{YY} & k_{ZY}\\ k_{XZ} & k_{YZ} & k_{ZZ} \end{array}\right]\cdot \left[\begin{array}{c} E_{X}\\ E_{Y}\\ E_{Z} \end{array}\right]$$
where $\left[\begin{array}{ccc} k_{XX} & k_{YX} & k_{ZX}\\ k_{XY} & k_{YY} & k_{ZY}\\ k_{XZ} & k_{YZ} & k_{ZZ} \end{array}\right]$ is a sensitivity coefficient matrix, which uniquely determines the relationship between the electric-field intensity to be measured and the output voltage of the induction motor.

In the actual calibration process, the three electric-field components, $\left(E_{X},E_{Y},E_{Z}\right)$ need to be demodulated by using the output voltage signals of the sensor’s induction electrodes. Thus, the 3D electric-field intensity can be measured. The decoupled calculation formula of the electric-field components to be measured can be obtained as follows:(10)$$\left[\begin{array}{c} E_{X}\\ E_{Y}\\ E_{Z} \end{array}\right]=\left[\begin{array}{ccc} c_{XX} & c_{YX} & c_{ZX}\\ c_{XY} & c_{YY} & c_{ZY}\\ c_{XZ} & c_{YZ} & c_{ZZ} \end{array}\right]\cdot \left[\begin{array}{c} u_{X}\\ u_{Y}\\ u_{Z} \end{array}\right]$$
where $\left[\begin{array}{ccc} c_{XX} & c_{YX} & c_{ZX}\\ c_{XY} & c_{YY} & c_{ZY}\\ c_{XZ} & c_{YZ} & c_{ZZ} \end{array}\right]$ is the decoupled calibration matrix C of the 3D electric-field sensor.

The precision of matrix C determines the accuracy of the electric-field intensity to be measured, which is demodulated by the sensor output signals. Therefore, to minimize the coupling interference and realize the accurate measurement of each component of the 3D electric field, a decoupled calibration is needed for the 3D electric-field sensor. The nature of the calibration is to solve the optimal decoupled calibration matrix C based on the known output voltage data of the electric field and sensor.

## 3. Decoupled Calibration Method Based on Multi-Output Support Vector Regression (SVR)

### 3.1. SVR Model

The 3D electric-field sensor is a multi-input and multi-output system. Based on the input and output models of the 3D electric-field sensor proposed in Figure 3, a multi-output SVR model was constructed, as shown in Figure 4, where *x* and *y* are the input and output of the training data set, respectively, and m and n are the dimensions of the input and output, respectively. In this model, the training data set was divided into n subspaces according to the dimensions. Each subspace was composed of m-dimensional inputs and a 1D output. An SVR was constructed for each subspace to estimate the output characteristics of the subspace by regression. When the model was applied to the 3D electric-field sensor, *x* was input into the corresponding model of the voltage components of the three passages. The components of the electric-field intensity in the three directions in the Cartesian coordinate system corresponded to y, m = n = 3. The *f*(*x*) was shown as follows,
(11)$$f(x)=\sum_{k=1}^{3}f^{k}(x)$$

The combination of the SVR components in the three subspaces were the full-range output characteristics of the sensor.

### 3.2. ν-SVR Model

A typical SVR algorithm is the standard ε-SVR algorithm [15]; however, the parameter ε in its insensitive loss function cannot be calculated automatically. Therefore, in this paper, the ν-SVR algorithm [16] was used to replace the empirical error ε with a quantitatively significant parameter ν to improve the adaptability of the algorithm. In this study, ν-SVR was used to decouple and correct the measured values of the 3D electric-field sensor. Through kernel function mapping, a linear regression function in the high-dimensional space was constructed, as follows:(12)f(x)=ωϕ(x)+d
where ω is the weight, and d is the intercept. The calculation procedure to determine ω, d, and f(x) is as follows.

(1) The training set is selected, which is denoted as $T=\left\{\left(x_{1},y_{1}),(x_{2},y_{2})\cdots ,(x_{l},y_{l}\right)\right\}$, where $x_{i}\in R^{m}$ is the input, $y_{i}\in R^{n}$ is the output, and i=1,2,3,⋯,l. The parameter *l* is the number of training sample points, $f^{k}(x)$ is the regression function, and k=1,⋯,a, where *a* and *b* are the dimensions of the input and output variables, respectively.

(2) An appropriate number of support vectors ν, the penalty factor C, and an appropriate kernel function $K(x_{i},x_{j})=\phi (x_{i})\phi (x_{j})$ are selected.

(3) The following optimization problem is solved:(13)$$\begin{array}{l} \min\sum_{i=1}^{l}(\alpha_{i}^{*}-\alpha_{i})y_{i}+\frac{1}{2}\sum_{i,j=1}^{l}\left[\left(\alpha_{i}^{*}-\alpha_{i})(\alpha_{j}^{*}-\alpha_{j})K(x_{i},x_{j}\right)\right];\\ s.t.\sum_{i,j=1}^{l}(\alpha_{i}^{*}-\alpha_{i})=0,\sum_{i=1}^{l}(\alpha_{i}^{*}+\alpha_{i})\leq Cv \end{array}$$
where $\alpha_{i}^{*},\;\alpha_{i}\in \left[0,\frac{C}{l}\right];\;i=1,2,\cdots ,l$.

The optimal solution is represented as follows:(14)$$\bar{\alpha }={\left[{\bar{\alpha }}_{1}\;{\bar{\alpha }}_{1}^{*}\;\cdots \;{\bar{\alpha }}_{l}\;{\bar{\alpha }}_{l}^{*}\right]}^{T}$$

(4) The regression parameters ω and d are determined with weights, as follows:(15)$$\omega =\sum_{i=1}^{l}({\bar{\alpha }}_{i}^{*}-{\bar{\alpha }}_{i})\phi (x_{i})$$

Two components ${\bar{\alpha }}_{j},{\bar{\alpha }}_{k}^{*}$ in the open interval (0,C/l) and their corresponding support vectors $\left[\begin{array}{cc} x_{j} & y_{j} \end{array}\right]$ and $\left[\begin{array}{cc} x_{k} & y_{k} \end{array}\right]$ are selected, where the intercept is as follows:(16)$$d=\frac{1}{2}\left\{y_{j}+y_{k}-\left[\sum_{i=1}^{l}\left({\bar{\alpha }}_{i}^{*}-{\bar{\alpha }}_{i}\right)K(x_{i},x_{j})+\sum_{i=1}^{l}\left({\bar{\alpha }}_{i}^{*}-{\bar{\alpha }}_{i}\right)K(x_{i},x_{k})\right]\right\}$$

(5) A regression decision function is constructed, as follows:(17)$$f(x)=\sum_{i=1}^{l}({\bar{\alpha }}_{i}^{*}-{\bar{\alpha }}_{i})K(x_{i},x)+d$$

The regression bias, which is treated as the solution of the optimization problem, is as follows:(18)$$\varepsilon =\sum_{i=1}^{l}({\bar{\alpha }}_{i}^{*}-{\bar{\alpha }}_{i})K(x_{i},x_{j})+d-y_{j}$$

## 4. Calibration Devices and Experiment Methods

### 4.1. Calibration Device

In this study, the method of spatially oriented decomposition of a 3D electric field was studied, and a computational model for the theoretical value of the electric field was established. A measurement calibration device was also designed, which could measure the direction angle between the sensor’s electrode and the 3D electric field. The 3D model of the calibration device is shown in Figure 5. The device was composed of a 20 kV high-voltage power supply, an electric-field box, a rotation table to fix the electric-field sensor, a flat-plate oscilloscope, and a computer. The high-voltage power supply was a GLOW28720 high-voltage electrostatic generator with a basic direct current (DC) voltage output of 0–20 kV. The oscilloscope was a TO1104 four-channel flat-plate oscilloscope produced by the Micsig Company. The output signal of the oscilloscope was not interfered with by the power frequency, and the analog bandwidth was 100 MHz. The three-channel analog output signals from a signal amplification circuit of the electric-field sensor were detected by a high-voltage probe connected to the oscilloscope. By adjusting the voltage U of the high-voltage power supply, an approximately uniform electric field was generated between the two parallel aluminum plates.

### 4.2. Measurement of Coupling Coefficient between Poles of 3D Electric-Field Sensor

To ensure the accuracy of the decoupled calibration algorithm, a standard uniform electric field was generated using an electric-field box. The induced voltage generated by the induction electrode of the 3D electric-field sensor that rotated in the uniform electric field was measured. The inter-pole coupling coefficient was measured by testing the amplitudes of the induced voltage signals in different directions.

A 3D electric-field sensor containing three mutually orthogonal distributed induction electrodes was selected. Since the sensor only had three sensitive units, its decoupled calibration matrix C was a 3 × 3 matrix. A local coordinate system was established based on the sensor, and the *X*-, *Y*-, and *Z*-axes of the local coordinate system were in the directions of the sensor electrodes S1, S2, and S3, respectively. One-dimensional electric field was applied by electrostatic filed instrument, three groups of induction electrodes (*X*, *Y*, *Z*) of the three-dimensional electric field sensor are placed in the direction parallel to the one-dimensional electric field direction for three times respectively. By changing the DC voltage applied to the electrostatic filed instrument, the intensity of uniform electric field generated by the electric field meter was changed, the amplitudes of voltage generated by the three groups of induction electrodes were measured and obtained. These voltage data are applied to the traditional inverse matrix method and the method proposed in this paper, the decoupling calibration matrix was calculated. The output voltage amplitude of the induction electrode in *X*, *Y* and *Z* directions obtained by three measurements are shown in Figure 6.

As shown in Figure 6a, when the induction electrode in the *X* direction of the three-dimensional electric field sensor is parallel to the electric field direction, the induction electrode in the *X* direction is greatly affected by the electric field, while the induction voltage amplitude generated by the induction electrode in the *Y* and *Z* directions is much smaller than that generated by the induction electrode in the *X* direction. In fact, since the applied electric field is parallel to the *X* direction, and the theoretical value of the electric field intensity in the *Y* and *Z* directions is 0, the voltage generated by the sensor in the *Y* and *Z* directions are essentially formed by the coupling effect of the electric field in the *X* direction on the *Y* and *Z* directions. At the same time, whether parallel or perpendicular to the direction of the electric field, when the applied voltage of the parallel plate is increased, the electric field intensity is increased, the amplitude of the induced voltage signal is increased, and the relationship is approximately linear. Similarly, in Figure 6b,c, the *Y* and *Z* direction induction electrodes of the three-dimensional electric field sensor are placed parallel to the electric field direction respectively. The amplitude of the induced voltage parallel to the electric field is large, and that of the other two induced electrodes affected by the electric field coupling effect is small.

As shown in Figure 6, within the measurement range of electric field intensity, no matter in the parallel or vertical direction of the induction electrode, the amplitude of the output induced voltage is basically linear with the electric field value. When there is an induction electrode parallel to the electric field direction in *X*, *Y* and *Z* directions, the output voltage amplitude of the induction electrode is the largest, and the output voltage signal amplitude of the other two induction electrodes is lower. The relationship between the sensor input (electric-field intensity), the sensor output (voltage signals), and the sensitivity coefficient matrix obtained through fitting with Equation (9) is expressed as follows:(19)U=KE
where E is the 3 × l component matrix of the intensity of the electric field, l the number of training samples, U is the 3 × l voltage signal matrix, and K is the 3 × 3 sensitivity coefficient matrix. K is a constant matrix that can be obtained by the least-squares method, as follows:(20)$$K=U{\left(F^{T}F\right)}^{-1}F^{T}$$
when the 3D electric-field sensor was decoupled, the independent variables were the three voltage vectors output by the sensor. The three orthogonal electric-field intensity components of the three induction electrodes acting on the sensor can be expressed as follows:(21)E=CU

The decoupled calibration matrix can be expressed as follows:(22)$$C={(K^{T}K)}^{-1}K^{T}$$

The process of solving the decoupled calibration matrix C is actually the process of least-squares fitting of multivariate functions. To obtain a more accurate calibration matrix, the number of required calibration points should be much greater than the number of dimensions of the 3D electric-field sensor. With the traditional method, the following decoupled calibration matrix was obtained:(23)$$C=\left[\begin{array}{ccc} 36.3665 & -10.019 & -1.07461\\ -7.85491 & 37.9304 & -31.5856\\ -6.84204 & -11.9843 & 103.8711 \end{array}\right]$$

The calibration method for the data along the other two axes was similar to the method described above, i.e., only the position of the induction electrode of the electric-field sensor in the equations was changed. According to the three calibration equations, the amplitudes of the 3D electric-field components can be obtained as follows:(24)$$\left[\begin{array}{c} E_{x}\\ E_{y}\\ E_{z} \end{array}\right]=C\left[\begin{array}{c} U_{x}\\ U_{y}\\ U_{z} \end{array}\right]=\left[\begin{array}{ccc} 36.3665 & -10.019 & -1.07461\\ -7.85491 & 37.9304 & -31.5856\\ -6.84204 & -11.9843 & 103.8711 \end{array}\right]\cdot \left[\begin{array}{c} U_{x}\\ U_{y}\\ U_{z} \end{array}\right]$$

In the method proposed in this paper, the ν-SVR method described in Section 3.2 is used to calibrate the 3D electric-field sensor with the following five steps.

(1) The training and test sets are established. The inputs of the training and test sets are the amplitudes of the output voltage signals of the 3D electric-field sensor in three different directions, and the outputs are the electric-field intensities of the 3D electric field in the *X*-, *Y*-, and *Z*-directions.

(2) According to the output dimension of the model, the three-dimensional electric field sensor dimension is a three-input three-output model. The training data set is divided into three subspaces, and parameter optimization is conducted for each subspace. According to the training data set, the penalty factor, number of support vectors, and other parameters are obtained, and the kernel function K adopts the radial basis function. Regression decision function model was shown as Equation (25).
(25)$$\left\{\begin{array}{c} f^{1}(x)=\sum_{i=1}^{16}({\bar{\alpha }}_{i}^{*}-{\bar{\alpha }}_{i})\exp[\frac{-{\left\|x-x_{i}\right\|}^{2}}{2\sigma^{2}}]+d\\ f^{2}(x)=\sum_{i=1}^{16}({\bar{\alpha }}_{i}^{*}-{\bar{\alpha }}_{i})\exp[\frac{-{\left\|x-x_{i}\right\|}^{2}}{2\sigma^{2}}]+d\\ f^{3}(x)=\sum_{i=1}^{20}({\bar{\alpha }}_{i}^{*}-{\bar{\alpha }}_{i})\exp[\frac{-{\left\|x-x_{i}\right\|}^{2}}{2\sigma^{2}}]+d \end{array}\right.$$

(3) Regression analysis is conducted with the training data according to Section 3.2.

(4) The coefficient matrix is solved, and the test set data is assigned to the matrix. The coefficient matrix is solved, assigning the test set data to **P** and the prediction data to **R**. The regression matrix **Q** is solved according to **PQ = R**.

(5) According to the test data in Figure 7, the regression prediction method of SVR is adopted to obtain the decoupling calibration matrix, as shown in Equation (26).

(6) Regression model and algorithm evaluation were shown in Table 1.
(26)$$C=\left[\begin{array}{ccc} 35.5746 & -10.1878 & -1.07234\\ -8.1019 & 38.0331 & -30.8892\\ -6.8113 & -11.9843 & 104.9652 \end{array}\right]$$

Ideally, the coefficient matrix of the sensor is diagonal matrix. However, due to the coupling effect, according to Figure 6, the slope of the electric field amplitude curves of *X*, *Y* and *Z* in the three graphs is different. The induction coefficient of the induction electrode in each direction of the 3D electric field sensor is different. The induction coefficients in the *X* and *Y* directions are very close, and the induction coefficient in *Z* direction is quite different from that of *X* and *Y*. This is because the *X* and *Y* sensors have the same electrode structure, which are a quarter of a cylinder, but the z-direction sensor electrode is shaped like a quarter of circle. It is very different from *X* and *Y* directions, and the coupling coefficient is different from other directions. According to Equations (23) and (26), in the two different 3 × 3 matrixes, the theoretical values of matrix elements a11 and a22 should be the same, but because in the two different directions, the direct distance between the inductive electrode and the shielding electrode is different, and the thickness of the induction electrode is different. As a result, the values of a11 and a22 on the main diagonal of the matrix are different. This is related to the fabrication and installation error of the sensor’s induction electrode.

## 5. Analysis of Experimental Result

To test the validity of the proposed decoupled calibration method based on the SVR for the 3D electric-field sensor, the sensor measurement data were taken as the training data set. The decoupled algorithm based on the traditional least-squares method and the proposed method were both used, and the errors were compared.

A rotation table used for calibration was designed, as shown in Figure 7. The compass of the rotation table was marked with a scale to facilitate the adjustment of the rotation angle. To minimize the interference of the rotation table to the measured data, the rotation table was fixed outside the electric-field box, and an insulating wood board was connected to the electric-field box from the rotation table. The sensor was fixed on the wood board near the center of the electric-field box.

The method of solving the inverse matrix is referred to as Method A, and the proposed method is referred to as Method B. The measurement results of the two decoupled calibration matrices obtained by these two methods were compared.

A uniform intensity electric field of 12 kV/m was applied by electrostatic filed instrument, the electric field sensor is placed in a plane perpendicular to the *X* axis and rotated in the YOZ plane, *θ* is the rotation Angle, and the theoretical values of the electric field intensity in the *X*, *Y* and *Z* directions are shown in the Equation (27).
(27)$$\left\{\begin{array}{c} E_{xt}=0\\ E_{yt}=E\cdot \sin\theta \\ E_{yt}=E\cdot \cos\theta \end{array}\right.$$

According to Equation (27), the theoretical value of the electric field intensity in the *X* direction is 0, the electric field intensity in *Y* and *Z* directions was changed by the angle *θ* between the induction electrode and the actual electric field direction. With the rotation of the three-dimensional electric field sensor in the electric field changed, the rotation angle *θ* and output voltage data of the output sensor were recorded. The electric field intensity in different directions is calculated by voltage data and two decoupled calibration matrices obtained. When the electric field sensor rotated in a plane perpendicular to the *X* axis, the amplitude of the voltage generated by the three induction electrodes, and the obtained induced voltage amplitude are respectively used in the traditional inverse matrix (Method A) and the proposed method in this paper (method B). By substituting matrix (23) and matrix (26) into formula (24), the calculated values of the electric field components are obtained by calculation respectively. Figure 8 presents the calculated and theoretical values of the electric-field components obtained by the two different decoupled calibration matrices. The relative errors between the calculated value of the total electric field and the theoretical value obtained by the two decoupled calibration matrices are shown in Figure 9.

Figure 8b,c show that the theoretical values obtained by the two methods had good consistency in the *Y*- and *Z*-directions, and the two methods both could effectively eliminate the coupling interference. According to Equation (27), as the electric field direction is perpendicular to the *X* direction, the sensor rotates in the YOZ plane, the theoretical value of electric field intensity in the *X* direction is 0. Therefore, in Figure 9a, since the theoretical value of electric field in the *X* direction is 0. In the *X*, *Y*, and *Z* directions, the decoupling calibration matrix obtained by the proposed method in this paper, compared with the traditional least square solution to obtain the decoupling matrix, the electric-field component calculated is closer to the theoretical value of electric field intensity.

The relative errors between the calculated value of the total electric field and the theoretical value obtained by the two decoupled calibration matrices are shown in Figure 9. The relative error given by the proposed method was smaller than that of the traditional method. As shown in Figure 9c, the relative error given by the traditional method was affected by the angle *θ*, with the largest errors at *θ* = 0 and *θ* = π/2 and the smaller errors at *θ* ≈ π/4. In contrast, the relative errors of the proposed method were less affected by *θ*. Therefore, the proposed method can effectively avoid the errors caused by the angle between the electric-field direction and the direction perpendicular to the induction electrode during the measurement process.

The intensity of the three-dimensional total electric field is calculated by the Equation (28).
(28)$$\left|E\right|=\sqrt{E_{x}^{2}+E_{y}^{2}+E_{z}^{2}}$$

The actual amplitude of the total electric field is 12 kV/m, the calculated and theoretical values of the total electric field obtained by the two decoupled calibration matrices are shown in Figure 10, and their relative errors are shown in Figure 11.

The theoretical values of the electric-field components obtained by the two decoupled calibration methods both could effectively eliminate the coupling interference after synthesis. However, the relative errors between the calculated and theoretical values of the electric-field intensity obtained by the proposed method were smaller and were not affected by the rotation angle *θ*. Table 2 shows the relative error between the calculated value of electric field and the theoretical value obtained by two decoupling calibration methods. The maximum relative errors were reduced from 16.3% to 5.87% and the average relative errors reduced from 4.58% to 2.72%. The experimental data showed that the proposed method based on the multi-output SVR algorithm could effectively eliminate the coupling interference of the electric-field components, improve the calibration accuracy of the electric-field sensor, and achieve accurate measurement of the 3D electric field.

## 6. Conclusions

The decoupled calibration method for the 3D electric-field sensor based on the multi-output SVR was proposed to solve the measurement accuracy problem of the electric-field intensity caused by the coupling interference of the electric-field sensor given by the 3D electric-field components. The proposed method could replace the traditional method of solving the inverse matrix using the least-squares method. The decoupled calibration process for the 3D electric-field sensor was designed. A measurement calibration device was constructed to realize the angle between the sensor and the electric-field vector. The computational model of the theoretical electric-field value was established to obtain the 3D decoupled calibration matrix. The following conclusions were drawn.

(1) Although the traditional decoupled calibration method can reduce the coupling interference by inverse matrix computation, the calculation errors are large, and it can easily miss the optimal solution due to the complex computations of the inverse matrix. Moreover, the traditional method can easily generate an ill-conditioned matrix. All these problems lead to large errors and cannot satisfy the measurement accuracy requirements of the 3D electric-field intensity. This paper proposes a decoupled calibration method to directly search for the global optimal solution in the solution space. The proposed method can avoid the ill-conditioned matrix and hence has high reliability and good decoupling accuracy.

(2) The test data of the electric-field intensity components in the *X*-, *Y*-, and *Z*-directions given by the two decoupled calibration methods were compared. The test results showed that the proposed method yielded better consistency between the calculated and theoretical values than the traditional method.

(3) The test data of the total electric-field intensity given by the two decoupled calibration methods were compared. The test results showed that compared to the traditional method, the proposed method reduced the maximum relative error from 16.3% to 4.58% and reduced the average relative error from 5.87% to 2.72%. Such reductions effectively reduced the coupling interference of the 3D electric-field components to the electric-field sensor, improved the calibration accuracy of the 3D electric-field sensor, and achieved accurate measurement of the 3D electric-field intensity.

(4) Ideally, the coefficient matrix of sensor is diagonal matrix. However, due to the coupling effect, non-main-diagonal elements of the matrix are not 0. The structure of induction electrode is the same in *X* and *Y* directions, which leading to the same elements a11 and a22 of coupling coefficient matrix in theory, but because in the two different directions, the direct distance between the inductive electrode and the shielding electrode is different, and the thickness of the induction electrode is different, there are differences between these two elements. Through calculation and analysis of coupling coefficient matrix, necessary data support is provided for the manufacturing, installation and calibration of sensor induction electrode.

## Figures and Tables

**Figure 1 sensors-21-08196-f001:**
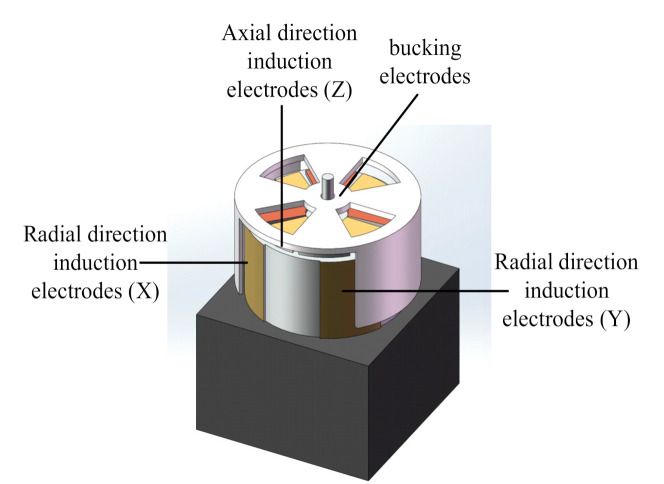
Model of field mill type three dimensional electric field sensor.

**Figure 2 sensors-21-08196-f002:**
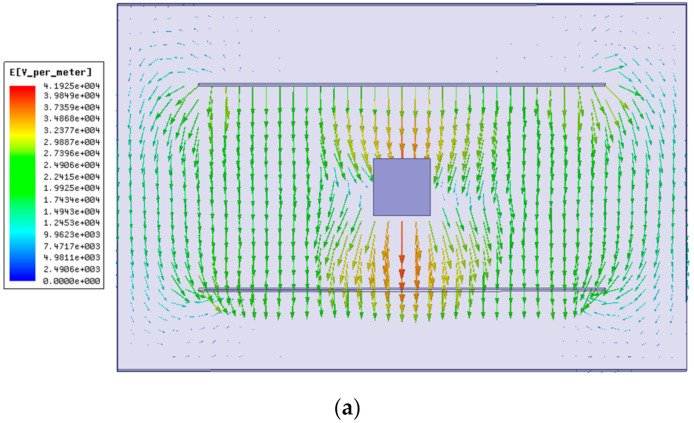
Distribution of eletronic lines and potential cloud. (**a**) Distribution of electronic lines; (**b**) Potential cloud.

**Figure 3 sensors-21-08196-f003:**
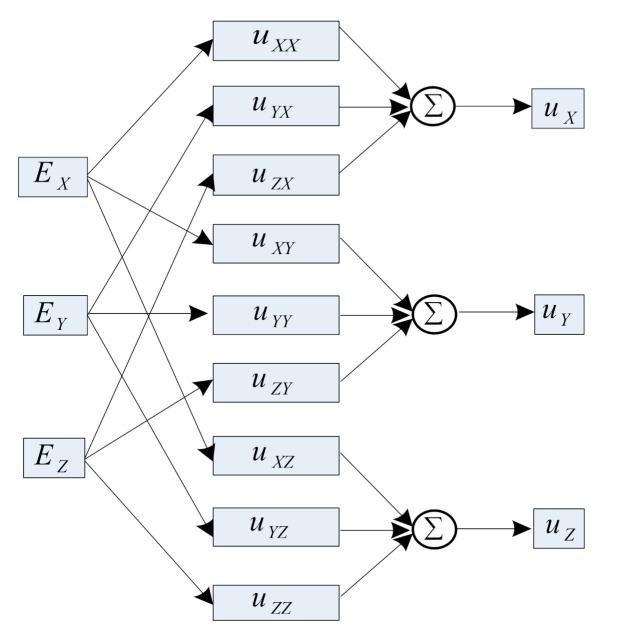
Input and output model of three dimensional electric field sensor.

**Figure 4 sensors-21-08196-f004:**
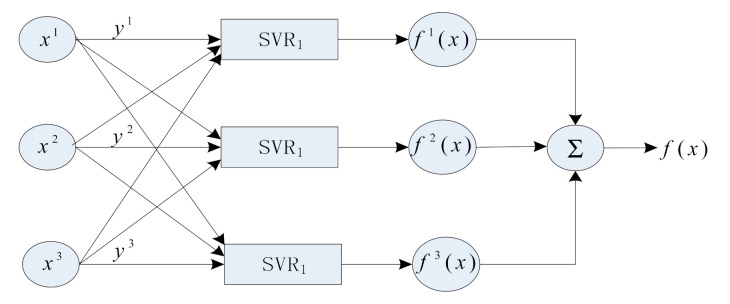
Multi-output support vector regression model of three dimensional electric field sensor.

**Figure 5 sensors-21-08196-f005:**
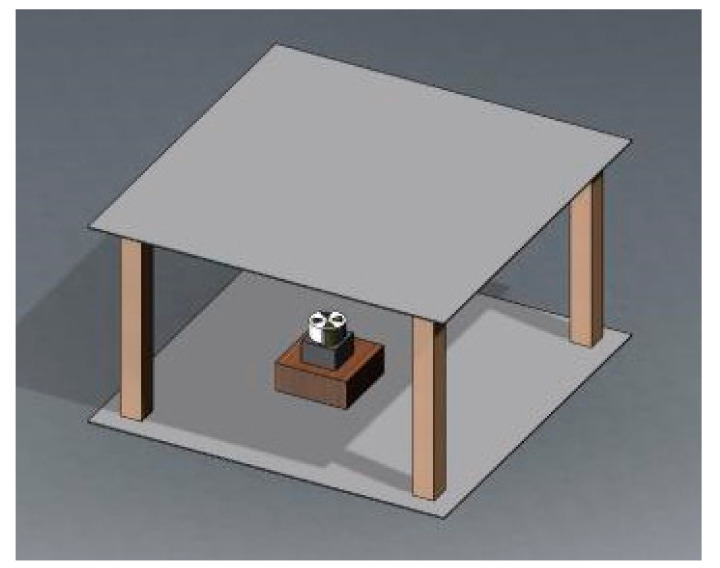
Calibration device model of three-dimensional electric field sensor.

**Figure 6 sensors-21-08196-f006:**
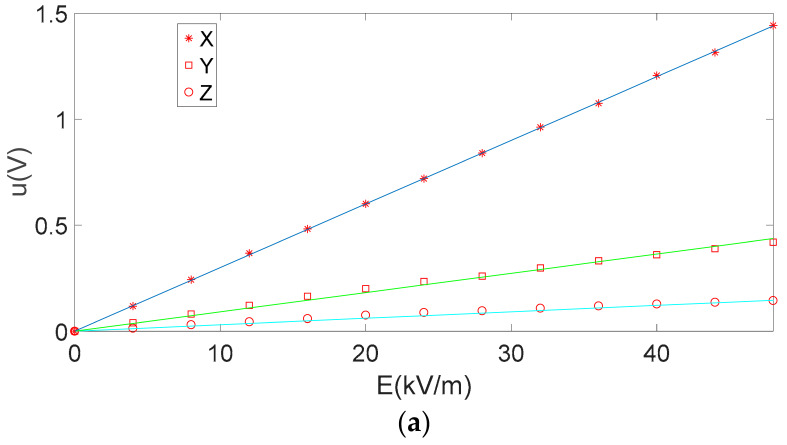
The amplitude of the induction electrodes output voltage; (**a**) *X* axis direction; (**b**) *Y* axis direction; (**c**) *Z* axis direction.

**Figure 7 sensors-21-08196-f007:**
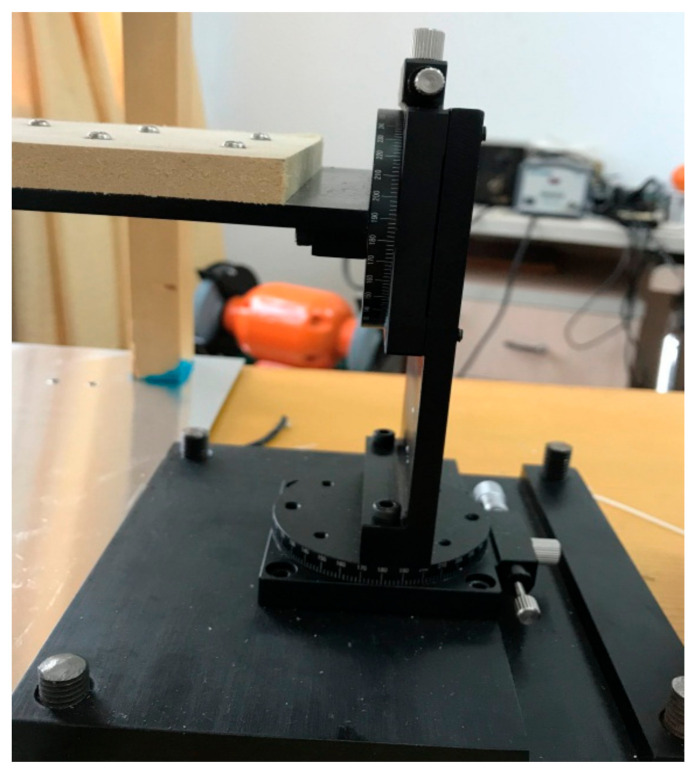
Rotation table.

**Figure 8 sensors-21-08196-f008:**
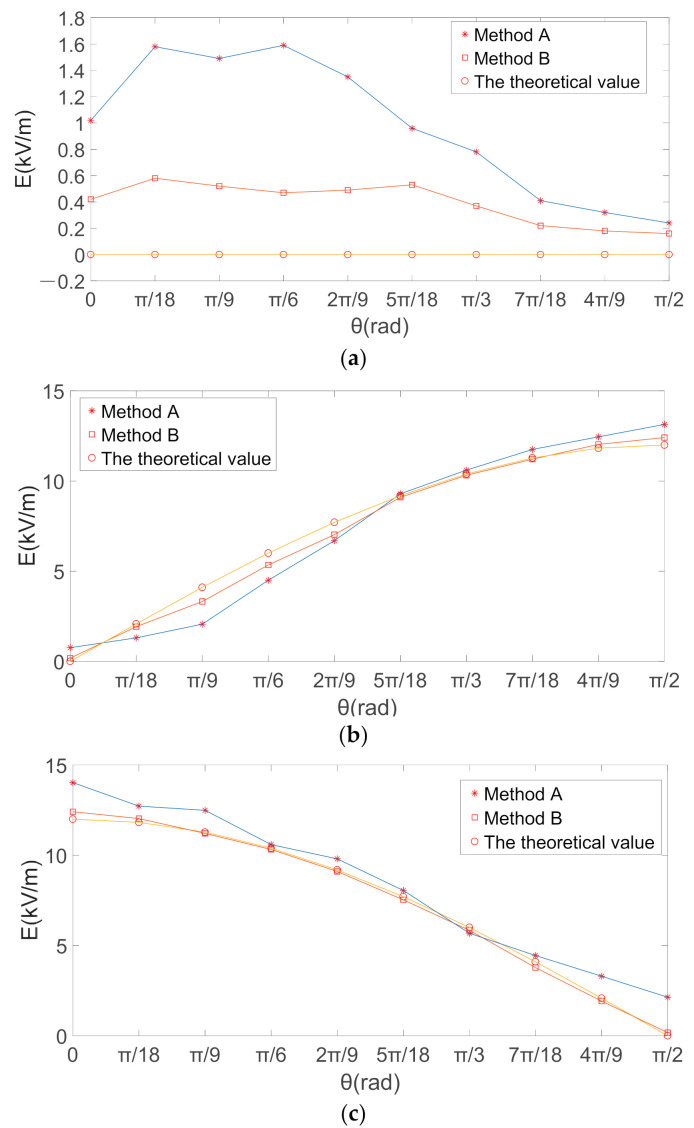
Electric-field intensity, E (kV/m). (**a**) *X* axis direction; (**b**) *Y* axis direction; (**c**) *Z* axis direction.

**Figure 9 sensors-21-08196-f009:**
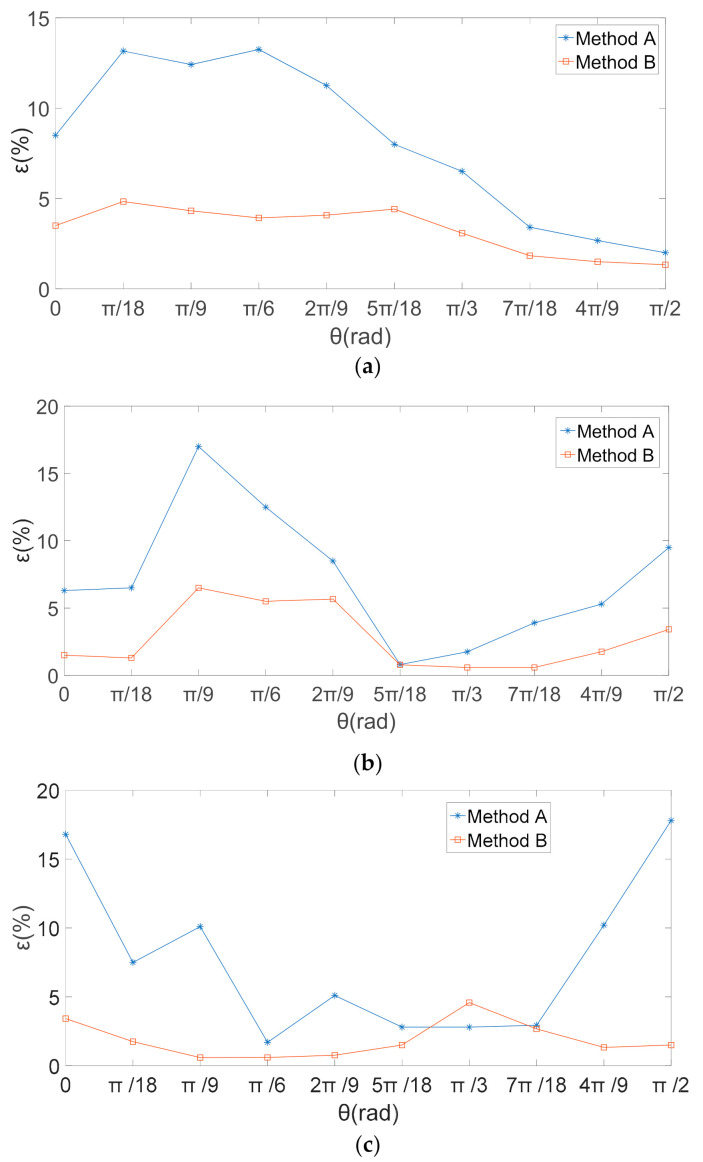
Relative Error in Electric field intensity (ε/%). (**a**) *X* axis direction; (**b**) *Y* axis direction; (**c**) *Z* axis direction.

**Figure 10 sensors-21-08196-f010:**
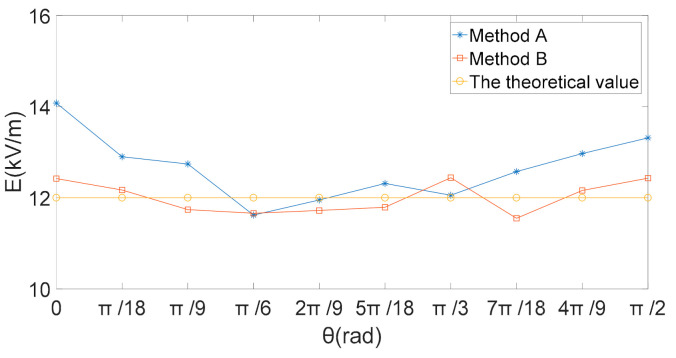
Electric field intensity (E/(kV/m)).

**Figure 11 sensors-21-08196-f011:**
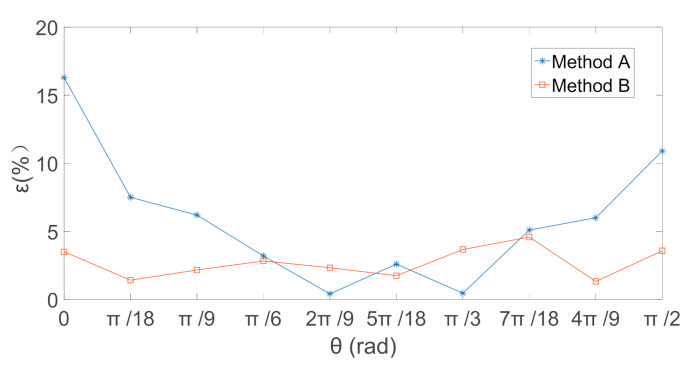
Relative Error in Electric field intensity (ε/%).

**Table 1 sensors-21-08196-t001:** Regression model and algorithm evaluation.

Regression Decision Function Model	Optimum Penalty Factors *C*	Number of Support Vectors	Constant *d*	Mean Square Error	Square Correlation Coefficient
$f^{1}(x)$	8	16	0.0035	6.33 × 10^−5^	0.994
$f^{2}(x)$	8	16	0.0074	6.31 × 10^−5^	0.992
$f^{3}(x)$	16	20	0.0532	1.42 × 10^−4^	0.992

**Table 2 sensors-21-08196-t002:** The relative error between the calculated value of electric field and the theoretical value obtained by two decoupling calibration methods.

Electric Field Intensity	Traditional Least Squares Method for Solving Inverse Matrix		Method Proposed in This Paper	
	Maximum Relative Error	Mean Relative Error	Maximum Relative Error	Mean Relative Error
$E_{x}$	13.9%	8.2%	4.83%	3.27%
$E_{y}$	16.9%	7.21%	6.5%	2.76%
$E_{z}$	17.8%	7.77%	4.55%	1.88%
$E_{}$	16.3%	5.87%	4.58%	2.72%

## Data Availability

Not applicable.

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
