# Peer review of "A Decoupled Calibration Method Based on the Multi-Output Support Vector Regression Algorithm for Three-Dimensional Electric-Field Sensors"

_sensors, 2021, doi:10.3390/s21248196_

Round 1

Reviewer 1 Report

I appreciated you to give me an opportunity to read your interesting paper. However, your paper will be better than now, if you modify below contents. I am looking forward for your paper to being published in this journal.

1. In 22 lines, what is “he accuracy” mean?

2. I recommended you add more literature survey in introduction part.

3. Please, check the fitting line correct or not in figure 6 (a).

4. I cannot understand the theoretical value in figure 8 (a). Please, check this correct or not.

5. What reason were the differences between traditional method and your proposed method? If you write more these reason and discussion, your paper will be better.

Reviewer 2 Report

The presented manuscript paper proposes a method to avoid the coupling in the electric field measured. This manuscript has three parts, the first part consists in an introduction and theoretical background of the proposal method. Moreover, a linear optimization is applied using multioutput SVR. The second part, the calibration devices and experimental methods is explained, and, finally, the experimental results are yield in the third part. The optimization is obtained considering a significant parameter, I suppose that it is based as convex optimization. Later, they stablish the discussion and conclusions.

The main conclusion is that they have proposed a global optimal solution. The results yielded better than the traditional method and some statistical values are improved.

The paper has "flaw" which must be commented:

  • The linear optimization is not well presented. The problem looks a Quadratic Convex optimization. define convex and non-convex. It could be interesting specify what is this, when are convex and non-convex. What means. As well, talk about de Karush – Kuhn - Tucker conditions It is not clear which fundamentals are used to perform this optimization and why is global optimal optimization.
  • The regularization term is avoided. Normally, the parameters are evaluated in order to balance the bias and variance effects.
  • Which least-squares fitting of multivariate functions algorithm is used?
  • In general, the manuscript must deep into the optimization techniques and solve some typical issues. Why don’t they apply some cross-validation technique for getting the generalized error?

In the following list, I report issues that should be addressed:

  1. Perform a full explain of the linear optimization or QCO, clarify.
  2. Establish the condition for robust optimization and relates with convex function.
  3. Try to solve the optimization with others optimization packages, yalmip as example.
  4. Which software package is used? share the code and data.
  5. Why do they use this optimization? are there others? Advantages and drawbacks. Has been Deep NN considered? Why? Why not?
  6. There are a lot of statistical evaluation data. What about R-adj?

Reviewer 3 Report

The title and contents of the paper should make it clear that your analysis is dealing with very high amplitude, DC, electric fields.  Normal atmospheric fields are of the order of 100V/m, and some readers will have interests in resolving electric fields of mV/m, such as AC electric fields from e.g. household wiring.  Are your results relevant to these potential readers?

3D induction electric field sensors as you illustrate have electric fields measured by rotation in two directions x,y, and by switching in the third z direction.  Have you accounted for different relative accuracy of the z direction as compared to the x-y directions, as well as different sensitivity?  This could be shown by error bars on your plots based on repeat measurements (Figures 8 to 10)

Round 2

Reviewer 1 Report

I appreciate you giving me a chance to read your revised manuscript. Thank you for your effort to revise your manuscript.

Reviewer 2 Report

The authors have improved the article with their new contributions.